# Mechanobiology and Primary Cilium in the Pathophysiology of Bone Marrow Myeloproliferative Diseases

**DOI:** 10.3390/ijms25168860

**Published:** 2024-08-14

**Authors:** Federica Tiberio, Anna Rita Daniela Coda, Domiziano Dario Tosi, Debora Luzi, Luca Polito, Arcangelo Liso, Wanda Lattanzi

**Affiliations:** 1Department of Life Science and Public Health, Università Cattolica del Sacro Cuore, 00168 Rome, Italy; federica.tiberio@unicatt.it (F.T.); domizianodario.tosi@unicatt.it (D.D.T.); luca.polito@unicatt.it (L.P.); 2Fondazione Policlinico Universitario A. Gemelli IRCCS, 00168 Rome, Italy; 3Department of Medical and Surgical Sciences, University of Foggia, 71122 Foggia, Italy; daniela.coda@unifg.it; 4S.C. Oncoematologia, Azienda Ospedaliera di Terni, 05100 Terni, Italy; d.luzi@aospterni.it; 5Department of Medicine and Surgery, University of Perugia, 06129 Perugia, Italy

**Keywords:** myeloproliferative neoplasms, mechanobiology, bone marrow niche, primary cilium, hematopoietic stem cells, mesenchymal stromal cells

## Abstract

Philadelphia-Negative Myeloproliferative neoplasms (MPNs) are a diverse group of blood cancers leading to excessive production of mature blood cells. These chronic diseases, including polycythemia vera (PV), essential thrombocythemia (ET), and primary myelofibrosis (PMF), can significantly impact patient quality of life and are still incurable in the vast majority of the cases. This review examines the mechanobiology within a bone marrow niche, emphasizing the role of mechanical cues and the primary cilium in the pathophysiology of MPNs. It discusses the influence of extracellular matrix components, cell-cell and cell-matrix interactions, and mechanosensitive structures on hematopoietic stem cell (HSC) behavior and disease progression. Additionally, the potential implications of the primary cilium as a chemo- and mechanosensory organelle in bone marrow cells are explored, highlighting its involvement in signaling pathways crucial for hematopoietic regulation. This review proposes future research directions to better understand the dysregulated bone marrow niche in MPNs and to identify novel therapeutic targets.

## 1. Introduction

Philadelphia-Negative Myeloproliferative neoplasms (MPNs) are a heterogeneous group of blood cancers characterized by genetic lesions affecting the stem cells of bone marrow. These conditions result in the overproduction of mature blood cells, including red blood cells, white blood cells, and platelets [1,2].

MPNs are chronic diseases that can manifest in various forms, the more frequent being polycythemia vera (PV), essential thrombocythemia (ET), and primary myelofibrosis (PMF), first described in 1951 by William Dameshek [3] (see Table 1 for classification of main MPNs). In particular, myelofibrosis (MF) can also develop following PV and ET (so called post-PV-MF and post-ET-MF). The classification of MPNs has been recently revised in order to identify different stages of the disease, i.e., “pre-fibrotic/early PMF”, which is a disease different from ET and is characterized by peculiar bone marrow (BM) morphology and is associated with the possibility of developing MF [4,5,6,7]. Specifically, PV is associated with an overproduction of progenitors and mature cells of the erythroid lineage; ET is characterized by an excessive production of megakaryocytes and platelets; PMF typically presents alongside bone marrow fibrosis and megakaryocytic lineage expansion [5]. In particular, in Appendix A are reported the updated diagnostic criteria according to the International Consensus Classification (ICC) and in Appendix A are reported the World Health Organization classification of hematolymphoid tumors (WHO-HEM5) [6,7].

In Europe, the incidence of MPN ranges between 0.4 and 2.8/100,000 in PV, 0.38 to 1.7/100,000 in ET, and 0.1 to 1/100,000 in PMF. Clinically, MPN are very heterogeneous neoplasms, characterized by a median survival of 20 years for ET, 14 years for PV, and 6 years for PMF; moreover, these values correspond to 33, 24, 15, respectively, when patients are less than 60 years old at the time of diagnosis. The clinical presentation of MPNs can vary widely, ranging from asymptomatic cases detected incidentally during routine blood tests to cases with severe symptoms such as fatigue, bleeding, and thrombosis, in particular for PV and ET, which can be associated with cardiovascular events [4].

Although MPNs are defined as chronic conditions, they can significantly impact patients’ quality of life and may progress to more advanced stages over time. Usually, from a clinical standpoint, the first abnormalities observed in patients are related to peripheral blood count. The values of hemoglobin concentration, mean cell volume (MCV), leucocytes, and platelets may range from normal to abnormal figures, and this information is also crucial to subsequently monitor the response to treatment and for disease progression. The diagnosis of MPNs typically involves a combination of clinical evaluation, peripheral blood tests, bone marrow biopsy, and genetic testing to confirm the presence of specific mutations [4].

The genetic basis of MPNs has been extensively studied, and several authors have demonstrated the presence of genetic mutations, particularly in *JAK2* (Janus kinase 2), *CALR* (calreticulin), and *MPL* (thrombopoietin receptor) genes, which play a pivotal role in driving the abnormal growth of blood cells [8,9,10,11]. For this reason, the expression “MPN phenotypic driver mutations” is used in reference to the abovementioned genes. The presence of an MPN driver mutation in the hematopoietic stem cells (HSCs) and myeloid progenitor cell compartment, in turn, drives cytokine-independent and cytokine-hypersensitive proliferative signals that cause an excessive production of mature blood cells from the myeloid lineage [12].

The bone marrow niche is clearly dysregulated on several levels in MPNs. Not only is the balance/proportion among different hematopoietic cell lineages is disrupted, but abnormalities in the extracellular matrix and in non-hematopoietic support cells are also described.

The extracellular matrix (ECM) physiologically supports the integrity and proliferation of the bone marrow niche. It consists of up to 300 protein components, which drive the maintenance of the HSC pool [13]. Mesenchymal stromal cells (MSCs) are key support elements in the bone marrow niche, where they interact with HSCs via paracrine and juxtacrine signaling and provide physical support to control hematopoietic stem and progenitor cells’ (HSPCs) maintenance and fate [14].

Notably, disruption in the architecture and cellular composition of the bone marrow associated with MPNs exposes HSCs to a profoundly distinct mechanical environment, with potential implications for progression of hematologic dysfunction and pathologies. 

The mechanical properties of the bone marrow microenvironment, including geometric properties and tissue architecture, matrix stiffness, blood flow in arteries and capillaries, and interstitial fluid flow, thus significantly impact HSC behavior [15]. In fact, numerous mechanosensors are present on hematopoietic and non-hematopoietic cells in the bone marrow, including integrins, mechanosensitive ion channels, and primary cilia [15].

In particular, primary cilia are single, non-motile, microtubule-based organelles projecting from the surface of most vertebrate cells, where they serve as pleiotropic chemo- and mechanosensors modulating proliferation, differentiation, and cell migration. Primary cilia are found on the majority of bone marrow cells [16]. They house an intense signaling, mostly including Hedgehog (Hh) and Wnt pathways. Given the key role played by Hh signaling in early hematopoietic development, the implication of the primary cilium in the development and homeostasis of the bone marrow is an intriguing research avenue poorly explored to date.

This review explores the role of mechanobiology in sensing external forces within the bone marrow niche and its impact on the development and function of HSCs and MSCs. We discuss mechanical cues related to cell structure, cell-cell and cell-matrix interactions, and mechano-responsive transcription factors. Additionally, we highlight the primary cilium and its potential implications in MPNs. In this respect, we propose future lines of research which will help increase our knowledge of the MPN-associated dysregulated niche and help understand how this niche can be successfully targeted in therapy.

## 2. Methods

### Search Strategy and Selection

Five assessors independently searched the PubMed database to identify relevant articles published up to June 2024. 

The search strategy was based on a selection of relevant keywords (Myeloproliferative neoplasms; mechanobiology; bone marrow niche; primary cilium; hematopoietic stem cells; mesenchymal stromal cells; and mechanical cues), word combinations of them, and their different spelling/plural terms. The first article selection was limited by selecting articles with one keyword in the title and a second keyword in all of the fields with the formula “(1° keyword[Title]) AND (2° keyword)” in the query box. A second tool-based screening was held with “Rayyan” by excluding duplicates, papers not containing article-related keywords in the abstracts or title, non-English, and non-full text available papers. No time frame was set for the publications date, but outdated articles were not considered. In the end, the remaining articles were read completely and papers with similar or duplicate information were discarded. Additionally, each assessor autonomously filled lacking information with papers that were prior excluded or that were not in the selections (Figure 1).

## 3. Mechanobiology of HSCs in the Bone Marrow Niche

The complex and delicate microenvironment surrounding HSCs in the bone marrow, known as the niche, can provide physical, chemical, and biological stimuli to regulate the survival, maintenance, proliferation, and differentiation of HSCs. Adult hematopoietic stem cells (HSCs) are localized in the bone marrow microenvironment, where two organized main cellular niches are recognized: the endosteal and the perivascular niche. As depicted on the left of Figure 2, the endosteal niche plays a key role in maintaining HSCs’ quiescence, while the perivascular niche (on the right), activates the cell cycle and promotes proliferation. Both biochemical and biomechanical factors contribute together to shape the environment. In particular, HSCs experience various mechanical forces like strain, pressure, and shear stress. These forces, both external and internal, are not separate but interact in a tensegrity model, where they influence each other. This interplay is believed to regulate hematopoiesis, HSC development, homeostasis, and malignancy, and the balance of HSCs in the body [18,19,20].

Hematopoietic cells feature diverse mechanosensors expressed as dedicated organelles and/or subcellular structures on their surface, such as integrins and focal adhesions, mechanosensitive ion channels, and primary cilia. In this paragraph, we provide an overview of the key mechanisms and structures involved in HSC mechanobiology, starting from the intracellular structures and moving outwards to the extracellular environment.

### 3.1. Intracellular Forces and the Role of the Cytoskeleton

Compared to mature blood cells, healthy bone marrow HSCs display a more rigid morphology, making them less prone to circulation and better retained within the marrow niche [21].

In leukemia cells, the lack of cytoskeletal mechanical tension weakens their adhesion to bone marrow niches, contributing to chemoresistance and persistent disease. This phenomenon accelerates leukemia progression and increases the likelihood of relapse [22].

Indeed, the cytoskeleton, made up of actin filaments, intermediate filaments, and microtubules, plays the leading role in transducing mechanical forces inside the cells. Cell-intrinsic forces are produced through actin filament polymerization and sliding along bipolar myosin filaments. Inhibiting myosin activity reduces the expansion of certain stem cells in response to changes in matrix stiffness. The strength of cell adhesion correlates with cytoskeletal contractility, indicating its importance in mechanical signal transmission between cells and their environment [23]. External mechanical stimuli can lead to cytoskeletal remodeling and tension rearrangement via various actin-binding proteins. Cells experiencing laminar shear stress develop “apical stress fibers” enriched with actin and myosin filaments, influenced by the cytoskeleton-related gene activation. Similar events occur during hematopoietic cell development, involving actomyosin activity controlled by myosin regulatory light chain 9 (MYL9) [24]. Researchers have also noted the upregulation of cytoskeletal proteins in response to mechanical forces. The cytoskeleton indeed regulates gene expression through the activation of mechano-responsive transcription factors. Yes-associated protein (YAP) and Transcriptional coactivators with PDZ-binding motif (TAZ) are well-known coactivators activated by mechanical stimuli. Their activity is typically limited to cells under mechanical stress [25]. These molecular mechanisms illustrate how mechanical conditions regulate hematopoietic stem cell actions [25]. Mechanosensing does impact several translational events, although all of the molecular pathways activated are still not known. The role of junctional interfaces is important to transmit internal forces, triggering cellular mechanical responses (Figure 3).

Interestingly, biomechanical signals generated in the cytoplasm are also ultimately transduced to the HSC’s nucleus that serves as a mechanosensitive organelle. This occurs both through the nuclear translocation of some key transcription factors governing cell differentiation fates and through direct nuclear mechanotransduction operated by the nuclear lamina, a meshwork of intermediate filament proteins named lamins [26,27] (Figure 3).

### 3.2. Cell-Cell and Cell-Substrate Interactions

Importantly, cells are not solitary entities, and they connect with neighboring cells and with the ECM through adhesion molecules, forming vital microenvironments relevant to HSCs’ fate [28]. Despite extensive research on cell-cell and cell-ECM interactions in bone marrow, a detailed understanding of adhesion force transmission in the bone marrow is still incomplete [29].

Transmembrane proteins involved in cell adhesive junctions, like cadherins and platelet endothelial cell adhesion molecule-1 (PECAM-1), are critical in HSC mechanosensing. Cadherins are known for their stability under mechanical forces and their sensitivity to substrate rigidity changes. PECAM-1, found at cell junctions, responds to fluid shear stress, transmitting signals to other molecules like VE-cadherin and VEGFR2 [30]. CD44, involved in lymphocyte trafficking and HPC homing, strengthens interactions under mechanical force, facilitating HSPC homing. Mechanical stimuli also influence adhesion reorganization in HSPCs, demonstrating their role in regulating adhesive junction dynamics [31]. Moreover, mechanical cues, like topography and forces, activate integrins, leading to the formation of mature focal adhesions (FAs), which help maintain cell mechanical balance [18].

Cell-ECM interactions allow for cells, including HSPCs, to sense and respond to mechanical signals in their surroundings. These contacts act both as sensors and transmitters of matrix elasticity [32]. Integrin-ligand adhesion, in particular, has been found to be critical for homing and anchoring HSCs and progenitors in bone marrow [15]. HSCs indeed interact with ECM components (e.g., fibronectin (FN), laminin, and collagen) through the binding of the extracellular domain of integrins with the conserved arginine-glycine-aspartic acid (RGD) motif shared by most ECM proteins. Different substrates can trigger mechanosensitive responses in HSCs via integrins, potentially regulating HSC behavior in response to matrix stiffness. Integrin αIIb (CD41), found in developing HSCs in the aorta-gonad-mesonephros (AGM) region, serves as a marker for nascent HSCs [33]. In the fetal liver (FL) hematopoietic niche, HSPC-ECM interactions depend on β1-integrin and on HSPCs binding to vitronectin and FN produced by hepatoblasts [28]. Furthermore, the recruitment of focal adhesion kinase (FAK) to FAs is crucial for transmitting external mechanical signals into cells. The ratio of phosphorylated FAK to total FAK, indicating FAK activity, increases significantly with substrate stiffness. In myoblasts, blocking FAK activity hinders stretch-induced alignment and differentiation [34]. FAK also activates the Rho family of small GTPases which relay signals from blood flow to YAP during embryonic HSPC production [35].

Cell-ECM interactions have a dual effect on mechanotransduction: matrix stiffness affects cell behavior, while cells exert traction forces on the ECM and secrete proteins that remodel the ECM, strengthening or degrading it and modifying adhesive interactions. ECM-remodeling proteins are crucial for controlling HSC quiescence, mobilization, and hematopoiesis in the HSC niche. Like the ECM diversity in other tissues, the structural and physical characteristics of HSC niches, such as stiffness and types of matrix ligands, vary significantly due to locoregional differences [36]. For instance, the endosteum region of the BM is stiff and rich in fibronectin (FN), while the perivascular space is softer with a high content of laminin [37]. The shapes of HSCs are linked to the stiffness of the matrix to which they are associated. On softer surfaces, HSCs tend to stay round, while on stiffer ones, they spread out more (Figure 3). Experiments with colony-forming units (CFUs) showed that stiffer substrates produced more multipotent CFUs compared to softer ones, although the type of matrix used played a role. Different coatings like FN, collagen, or laminin influenced the lineage fate of HSCs, affecting whether they developed into myeloid cells. Additionally, the elasticity of the substrate significantly impacts the expansion of HSPCs [38]. Adult HSCs in the bone marrow or circulating in blood vessels, as well as embryonic HSCs settling in specific areas during development, encounter several physical forces [39].

Studies on HSCs showed that the timing of blood flow onset is linked to the appearance of hematopoietic cells, suggesting that blood flow might influence HSC development locally within the “vascular niche” [40]. Recent data highlight the importance of circumferential strain, a component of blood flow, in HSC development, which seems to be conserved across species. Blood flow also helps organize actomyosin during the process of HSPC development and in their homing to the bone marrow. Additionally, during embryonic development, the dorsal aorta experiences compressive stresses from the surrounding tissues, while less is known about the mechanical forces in the fetal liver, another hematopoietic organ [41].

### 3.3. Vascular Cells as Transducers of Mechanical Signals to HSC in the Niche

Although adult HSCs in bone marrow might not directly experience blood flow, niche cells could indirectly influence them through paracrine signaling. For instance, endothelial cells and pericytes might be affected by fluid flow in bone blood vessels, impacting HSCs’ cycling and quiescence. Fluid flow in the bone’s canalicular network generates shear stresses, facilitating nutrient transport and mechanical activation of osteocytes [42]. HSCs have shown the ability to detect and respond to various mechanical cues. In addition to their unique ability to sense and respond to mechanical cues, endothelial cells exposed to force activate various well-known developmental pathways involved in regulating HSCs [43]. These pathways include PGE2, Wnt, Hedgehog, Notch, bone morphogenetic protein (BMP), and VEGF signaling. Shear stress triggers the formation of a mechanical sensor complex consisting of VEGF receptor 2 (VEGFR2 or FLK1), vascular endothelial (VE)-cadherin, and β-catenin. This suggests intricate interactions within the hematogenic endothelium and/or HSCs, forming a cooperative control system that influences the fate of HSCs [44]. It is also worth highlighting the role of nitric oxide (NO), produced from arginine by NO synthases (NOS), and having wide-ranging effects on cellular activities, including responding to mechanical signals. When blood flows, vascular endothelial cells release NO, which helps dilate blood vessels to regulate vascular tone [45]. Both fluid shear stress and mechanical stretches prompt rapid NO production. NO signaling regulates various processes in adult HSCs/HSPCs, but the effects differ depending on the source of the HSCs. For example, it can stimulate BM-HSC proliferation and myeloid differentiation while reducing their ability for long-term reconstitution [46].

## 4. The Role of Primary Cilium

The primary cilium (PC) is a slender antenna-like cell surface structure which was first observed over a century ago on renal tubule’s epithelial cells and was initially regarded as a non-functional vestigial structure [47]. Subsequent research has demonstrated that the PC is sustained by a microtubule-based axoneme, and research has also further illuminated its multifaceted role as a sensory antenna capable of detecting mechanical and chemical cues from the extracellular environment [48]. It then translates these signals into the activation of multiple intracellular molecular cascades, thereby orchestrating various processes crucial for development and maintaining homeostasis in all tissues [49]. A schematic view of the primary cilium structural organization is displayed in Figure 4.

The structural integrity of the primary cilia is paramount for their functionality, as evidenced by the wide and heterogeneous spectrum of diseases, collectively known as ciliopathies, associated with mutations that impair ciliary biogenesis, structure, and functions [50]. Many conditions, including arthritis, osteoporosis, polycystic kidney disease, heart failure, obesity, and cancer, have also been linked to primary cilia malfunctions [50].

### 4.1. Structural and Functional Overview

The PC axoneme extends outward from a basal body serving as an apical microtubule-organizing center located right below the plasma membrane (see Figure 4). This basal body originates from the mother centriole during the G0 phase or cell quiescence and consists of a ring comprising nine doublets of microtubules. Upon docking at the apical cell surface, the basal body defines cell polarity and initiates ciliogenesis [44]. The PC elongates from this docking site thanks to the elongation of a modified axoneme rooting from the basal body and forming the central shaft that supports the entire ciliary structure (Figure 4). As in motile cilia, the PC axoneme features a radial array of nine microtubule doublets (see Figure 4). Notably, the PC axoneme differs from that found in motile cilia by the absence of a central pair of microtubules and of dynein arms, resulting in a “9 + 0” structure devoid of motility. In contrast, motile cilia typically exhibit a “9 + 2” arrangement, featuring dynein arms that facilitate ciliary movement by interacting with the central pair. Although the ciliary membrane that lines the axoneme is continuous with the plasma membrane, it is partitioned from it through a specialized region at the base of the PC, called the transition zone (TZ). This is based on fibrous protein-based structures called Y-links or transition fibers (Figure 4), which physically separate the cytoplasm from the cilioplasm, regulating the export and import of molecules [51,52]. In addition, the ciliary membrane is enriched in sterols and sphingolipids [53], particularly at the base of PC, that provide this domain an increased elasticity and resistance to flow shear stresses [54]. The portion of the ciliary membrane that surrounds the base of the PC projects inside the cell, forming an invagination called the ciliary pocket (CiPo). This serves as a docking acceptor site for membrane-bound vesicles ensuring the trafficking of ciliary molecules and structural complements during the early stages of ciliogenesis [55,56,57].

To date, the PC’s role in mechanosensing and mechanotrasduction has been reported in a wide range of cell and tissue types [57]. Mechanosensing is initiated by the passive deflection of the primary cilium caused by mechanical stimulation (e.g., fluid flow or extracellular matrix stiffness) triggering biochemical responses mainly through Hedgehog and Wnt pathways. This process is modeled in four steps: firstly, the mechanical cue is presented to the cells; secondly, the PC catches the mechanical signal and bends; thirdly, the signal is transduced inside of the cell through the passage of second messengers thanks to channels on the ciliary membrane; and lastly, the transmitter activates a biochemical response [58]. One of the key biochemical pathways in these processes involves second messengers, which are pivotal in initiating intracellular signaling upon biophysical stimulus. Among these, the calcium ion (Ca^2+^) is a ubiquitous second messenger that regulates a plethora of signaling pathways and is also the key to PC mechanotransduction [59]. PC-mediated mechanosensing involvement in Ca^2+^ influx regulation was already shown in canine-derived kidney cells based in an in vitro model (MDCK) in which an extracellular fluid flow was mimicked in microfluidic devices to expose the PC membrane to shear stress and promote its bending [60]. Ca^2+^ intracellular accumulation was followed through a calcium-sensitive fluorescent probe by which the PC membrane was previously marked [60]. Recently, a human cell-based study demonstrated that a rapid intracellular increase in Ca^2+^ levels can occur as a result of a mechanical loading in potentially all human cell types, including kidney cells, osteocytes, osteoblasts, and neurons. This is due to the high-density presence on the PC of specific Stretch-Activated ion Channels (SAC), whose genetic defects have been correlated with numerous ciliopathies [61,62]. These calcium-dependent ion channels allow for the selective passage of Ca^2+^ following their activation by deformation in the PC membrane from different mechanical stimuli, which mainly include fluid flow and ECM stiffness changes [63]. In particular, studies have shown that using a parallel plate flow chamber to mimic oscillatory fluid shear flow at a frequency of 1 Hz can greatly influence PC mechanosensing compared to a steady flow because it reflects the true kind of fluid flow in which PC acts [64,65]. Rapid changes in intracellular Ca^2+^ levels following fluid shear stress can trigger downstream transcriptional activity in bone cells, leading to significant cellular responses. In human MSCs, increases in intracellular Ca^2+^ induced by fluid shear stress are essential for gene expression modulation [66,67,68].

Strictly associated with intracellular calcium levels, recent research has expanded the understanding of mechanotransduction by identifying the primary cilium as a crucial cyclic adenosine 3′,5′-monophosphate (cAMP)-responsive mechanosensor in bone marrow MSCs. Fluid shear stress activates cAMP signaling within the primary cilium, which subsequently promotes osteogenesis through the intraflagellar transport protein 88 (IFT88). IFT88 is a key component of the intraflagellar transport complex B, necessary for primary cilium formation, and is transported anterogradely by kinesin [69]. In addition, downstream of the fluid shear-dependent increase in intracellular Ca^2+^, activation of cAMP signaling seems mediated by adenylyl cyclase 6 (AC6), a specific localized component of the ciliary microdomain. An in vivo study using an AC6 murine knockout model demonstrated that the loss of AC6 in leptin receptor-expressing stromal cells attenuates loading-induced endosteal bone formation, highlighting the main role of AC6 in the primary cilium mechano-signaling [70].

### 4.2. Ciliary Signaling and Their Relevance in HSC Biology

The main signaling cascades involved in intracellular ciliary transduction are Hh, Wnt, and YAP/TAZ pathways. 

The Hh signaling pathway is activated by the chemosensing activity of PC. The Patched1 receptor (PTCH1) is localized to the ciliary membrane, facing the outside; when at rest (i.e., in the absence of ligand binding), it inactivates and excludes the transmembrane protein Smoothened (SMO) from the PC. When a Hh ligand binds to PTCH1, this moves away from the cilium and relieves the inhibition of SMO, which in turn translocate to the PC [71,72]. Once activated, SMO triggers the dissociation of the suppressor of fused (SUFU) protein from the glioma-associated oncogene (GLI), which is then free to translocate to the nucleus via intraflagellar transport (IFT), regulating the expression of its target genes [72].

While the role of Hh signaling in the primary cilium is widely reported, the functional correlation of the Wnt pathway in the PC’s mediate transduction is still unclear. Canonical WNT/β-catenin signaling can differently regulate ciliogenesis [73]. In turn, the PC acts as an inhibitor for this signaling pathway as it sequesters Wnt signal effectors, such as disheveled and GSK3 [74]. Moreover, canonical Wnt signaling is inhibited by inversin, a key player of noncanonical Wnt signaling activation. The discovery of the functional localization of inversin at the PC basal body suggests a mechanism through which primary cilia promotes the activation of noncanonical Wnt signaling via inhibiting the (canonical) Wnt/β-catenin pathway [75]. Indeed, downregulation of canonical Wnt activation increases the degradation of β-catenin by proteasome [76]. Most interestingly, several primary cilia-related genes correlate with the upregulation of canonical Wnt signaling when depleted. Among these, the lowered expression of KIF3A and BBS4 strongly relates to increased β-catenin expression, while BBS4 interaction with the proteasome 26S subunit, non-ATPase 10, promotes the recruitment of proteasomes nearby the centrosome [77,78,79]. To date, it remains controversial how Wnt signaling regulation is determined by the primary cilia. Although the primary cilia-mediated transduction and downstream effects of Wnt stimulation appear mutually influenced, all collected observations suggest a main inhibitory role of PC against Wnt signaling via the degradation of β-catenin.

In addition, YAP can be activated by various mechanical cues like substrate stiffness, cyclic stretch, and shear stress, leading to its movement from the cytoplasm to the nucleus. YAP also plays a role in controlling tissue tension mediated by actomyosin during mechano-morphogenetic processes [80]. Goode et al. [81] observed the precise timing of YAP’s movement into the nucleus in endothelial cells just before the process of hematopoietic development in mice. Functional experiments confirmed that the interaction between TEAD and YAP regulates early hematopoietic specification during specific developmental stages [81]. Importantly, a wealth of data suggests a close connection between the primary cilium and some of the molecular pathways such as that of YAP/TAZ. Several authors have shown that mechanical signals from the primary cilium can influence the localization and activity of YAP/TAZ, thereby modulating gene expression and cellular behavior. The YAP/TAZ signaling pathway, in turn, has been implicated in the regulation of ciliogenesis and can influence the expression of genes involved in ciliary assembly and maintenance [82].

### 4.3. The Functional Significance of the Primary Cilium in the Bone Marrow Niche

Primary cilia are widely expressed in bone marrow cellular populations, including endothelial cells, hematopoietic progenitors, and most human blood and stromal support cells [16]. Recent evidence shows that the primary cilia play a crucial role in decoding the mechanical and chemical signals of the cellular microenvironment [83].

The PC’s role in mechanotransduction is pivotal for maintaining HSC quiescence, proliferation, and differentiation, enabling HSCs to respond dynamically to changes in their physical environment, thereby influencing their fate and function.

In zebrafish embryos, the structure of the primary cilia and their role in Notch signaling were observed in the region where HSPCs develop [84]. Dysfunction of the primary cilia led to severe defects in HSPCs, which could be partially rescued by overexpressing a component of the Notch signaling pathway. Moreover, genes involved in Notch signaling were affected in embryos lacking blood flow, suggesting a connection between blood flow, the primary cilia, and HSPC development. These findings strongly suggest that the primary cilia play a role in sensing and transmitting mechanical signals, although their precise role in HSCs is yet to be fully elucidated [85].

Additionally, several authors have shown that vascular endothelial cells rely on the primary cilia to sense fluid flow, which in turn regulates the production of NO, a key modulator of HSC activity. Since HSCs are often located near blood vessels in bone marrow, it is possible that mechanical stimulation of NO production via the primary cilia could further influence HSC behavior, although this mechanism requires further investigation [86].

Moreover, bone marrow MSCs represent a subset of nonhematopoietic cells located within the bone marrow and are endowed with multilineage potential along with extensive clonogenic properties maintained in the culture [87,88]. MSCs use the primary cilia to detect mechanical cues in their environment, which are necessary for their response to bone-forming signals and controlled growth [89,90,91]. When this mechanosensing structure is disrupted, the ability of mechanical signals to promote bone formation is compromised. Both osteoblastic cells and MSCs are crucial elements of bone marrow environments that support HSCs [92].

MSCs play multiple roles in bone marrow homeostasis. They MSCs contribute to bone formation and remodeling by differentiating towards the osteogenic lineage. They also regulate osteoclast activity through the secretion of factors such as osteoprotegerin (OPG) and the receptor activator of nuclear factor kappa-B ligand (RANKL), thereby influencing bone resorption. MSCs also exhibit valuable immunomodulatory properties, suppressing immune responses and promoting tolerance. They can inhibit the proliferation and function of various immune cells, including T cells, B cells, natural killer cells, and dendritic cells through cell-cell contact and secretion of soluble factors [93,94,95,96]. Finally, MSCs provide crucial support for HSCs by regulating their self-renewal, differentiation, and homing within the bone marrow niche through paracrine and juxtracrine signaling within the BM niche environment. They secrete various cytokines, growth factors, and extracellular matrix proteins that create a supportive microenvironment for HSC maintenance and proliferation [97]. In response to tissue damage, MSCs secrete a diverse array of ECM constituents, paracrine factors, and extracellular vesicles (EVs; predominantly exosomes). Upon activation, MSCs can also augment their own population and replenish specific constituents of the BM niche through the differentiation or attraction of supportive cells to a niche. These actions of MSCs are directed, either directly or indirectly, towards the preservation of resident stem cells following tissue injury [98]. 

A significant contribution of MSCs to tissue regeneration is believed to be achieved due to their ability to respond to mechanical stimuli from the microenvironment with high plasticity.

Yet, despite the advancements in understanding MSC biology, the entire array of regulatory mechanisms through which MSCs maintain their stemness and regulate HSC homeostasis within the BM niche microenvironment are still yet to be determined [14,99]. It has indeed been demonstrated that bone marrow-derived MSCs also express PCs, which play pivotal roles in regulating their properties and fate [89]. The PCs in MSCs contribute to intercellular communication through sensing signals from the ECM and thus regulating developmental processes and morphogenic gradient sensing [48,69,91]. Several studies described that PC in MSCs serve to respond to microenvironmental factors, sensing both chemical and physical extracellular cues, coming from niche specific conditions [91]. These signals play a crucial role in determining cellular responses, from adhesion and migration to the differentiation and regulation of gene expression within their niche [91]. Through the regulation of multiple signaling pathways housed in the cilioplasm, primary cilia act as regulators of MSC fate determination. Dysfunctions in PC formation or function have been linked to developmental disorders, highlighting the crucial role played in cell growth, survival, and differentiation [91,100]. Aberrant activations of key ciliary pathways as Hh, Wnt, and transforming growth factor beta (TGF-β) have also been shown to be critically involved in pathological myofibroblast transition and organ fibrosis [101].

In this context, recent genetic fate-tracing experiments and single-cell RNA-Seq data of the BM niche cell populations identified five major fibrosis-driving MSCs populations including GLI1^+^ myofibroblasts, Leptin receptor (LepR)-, platelet-derived growth factor receptor α (PDGFR-α)-, vascular cellular adhesion molecule 1 (VCAM1)-, and Nestin- expressing MSCs [102]. These contribute to MPN disorders affecting MSC and HSC pool maintenance within the bone marrow niche by the downregulation of HSC-supporting factors while increasing fibrogenic genes [102,103]. 

While the roles of primary cilia-dependent PDGF signals have been extensively demonstrated in fibroblasts, their significance in regulating cell proliferation and maintaining stemness in MSCs remains only partially explored. 

The activation of morphogen signaling pathways results in myofibroblast transition, but sustained morphogen signaling pathways lead to aberrant signaling and pathologic fibrosis [101]. Hh, Wnt, and other signaling pathways including TGF-β have been shown to also be critically involved in myofibroblast transformation and organ fibrosis and, remarkably, the receptors and key molecules involved in morphogen signaling pathways are localized in the primary cilium. Moreover, variations in oxygen tension markedly influence the formation and length of the cilia in primary bone marrow-derived MSCs, as chronic exposure to hypoxia exerts a repressive effect on Wnt and Hedgehog signaling pathways [101,104]. Interestingly, several recent studies highlighted a functional link between primary cilia and oxidative stress, demonstrating that oxidative stress can lead to the shortening and loss of primary cilia. Several Authors have shown that osteogenic potential loss of osteoblasts exposed to microgravity-induced oxidative stress was counteracted by promoting ciliogenesis and the maintenance of the normal length of the PC throughout treatment with cytochalasin D [105]. These findings emerged according to previous investigations which already connected the oxidative stress and primary cilia features. Han et al. [106] observed that cisplatin, a widely used anticancer drug, induced oxidative stress that resulted in the shortening of primary cilia in lung and kidney cells, contributing to lung damage and acute kidney injury. Similarly, another study reported that cigarette smoke extract impaired osteogenic differentiation in human mesenchymal stem cells by disrupting the distribution and integrity of the primary cilia through the generation of free radicals [107]. Additionally, the overactivation of TRPV4 (Transient Receptor Potential Cation Channel Subfamily V Member 4), a protein located throughout the primary cilia, has been associated with Ca^2+^ overload, mitochondrial dysfunction, and oxidative damage. The regulation of ciliogenesis, the formation of the primary cilia, is influenced by cellular stresses such as oxidative stress and exposure to agents like cisplatin. Dysfunction of the primary cilia, due to the loss of ciliary proteins like PCM1 (Pericentriolar Material 1) and TCTN3 (Tectonic Family Member 3), has been linked to increased apoptotic cell death in glioblastoma and neuronal cells in mice. Conversely, cilia-mediated Hedgehog (Hh) signaling, which relies on primary cilia, can activate autophagy, a protective mechanism that helps to eliminate damaged mitochondria under oxidative stress [108]. Moreover, a recent studio evidenced that the disruption of IFT88 expression, a key gene required for ciliogenesis, completely mocked the oxidative stress-induced molecular effects in rat calvarial osteoblasts, resulting in increased ROS (Reactive Oxygen Species) and downregulation of Superoxide Dismutase (SOD) and Catalase (CAT) enzyme activities [109]. This emerging reciprocal regulation between the primary cilia and oxidative stress underscores the adaptive role of ciliogenesis in mitigating mitochondrial stress and oxidative damage in mammalian cells.

Among the receptors localized on the ciliary membrane, TGF-β receptors mediate their own internalization from the surface of the primary cilium membrane through clathrin-mediated endocytosis. This internalization sparks phosphorylation and subsequent activation of transcription factors SMAD2/3 [110]. Once activated, SMADs assemble into a trimeric complex with SMAD4, which regulates the expression of specific profibrotic genes. Furthermore, the receptors for TGF-β on the primary cilia can initiate Hh signaling by SMO activation, leading to the conversion of full-length GLI into the transcriptional activator GLIA, which ultimately results in the expression of targeted profibrotic genes [101].

## 5. Altered Mechanobiology in MPN

Disruption in the architecture and cellular composition of the bone marrow occurring in myeloproliferative disorders exposes HSCs to a profoundly distinct mechanical environment, with significant implications for disease pathophysiology and the progression of hematologic dysfunctions. In particular, clonal myeloproliferative neoplasms caused by driver mutations in the *JAK2*, *MPL*, or *CALR* genes results in bone marrow fibrosis, increased the release of HSCs in systemic circulation, anemia, and inflammation. Bone marrow failure is observed with disease progression, being associated with osteosclerosis, extramedullary hematopoiesis in organs such as the spleen and liver, and splanchnic vein thrombosis [15]. These profound changes occur through a deep involvement of the bone marrow stroma as evidenced by myelofibrosis, neoangiogenesis, and osteosclerosis [111]. Indeed, MSCs have recently been identified as the main drivers of fibrosis in MPN. Specifically, Lepr^+^ MSCs undergo clonal expansion during PMF progression due to an increase in platelet-derived growth factor (PDGF) secreted by hyperplastic megakaryocytes [103]. This is accompanied by excessive reticulin and/or collagen fiber deposition in bone marrow ECM. Therefore, it has been speculated that altered niche stiffness associated with pathologic ECM deposition contributes to changes in mechanobiological signaling which reinforce MPN progression. On another note, Lee et al. have reported that *FOP* (*FGFR1 Oncogene Partner*), a gene involved in ciliogenesis, has also been shown to be a partner of the oncogenic fusion protein FOP-FGFR1 which causes a myeloproliferative neoplasm [112]. FOP is a centriolar satellite cargo; however, the functional significance of centrosome-kinase fusions in myeloproliferative neoplasms is not well understood, and it has been hypothesized that aberrant kinase localization is linked to the disease phenotype rather than to alterations in mechanotransduction.

In an intriguing previous study, we proposed that the IGFBP-6/SHH/TLR4 axis was implicated in alterations of the primary myelofibrosis microenvironment [113]. Our findings suggest that the IGFBP-6 protein may play a central role in activating the SHH pathway during the fibrotic process [113].

Also, the damage caused by chemotherapy and radiation used in bone marrow preconditioning regimens significantly models and changes the architecture and mechanobiology of the niche [114,115,116]. For instance, chemotherapy induces an imbalance in the proliferation/differentiation capabilities of MSCs, leading to reduced self-renewal and increased differentiation [117]. Of note, MSCs mechanical properties change during their differentiation along the adipogenic lineage (with reduced cytoskeletal density and overall cellular stiffness), which typically occurs following chemo/radiotherapy [118,119]. These changes ultimately hamper MSC paracrine and juxtacrine signaling that support HSC homeostasis in the niche. Therefore, the profound damage to the bone marrow stroma can negatively impact the function of radioresistant HSCs and progenitors, and thereafter the success of the hematopoietic graft.

## 6. Conclusions

This review provides a comprehensive exploration of the influence of machanobiology on both HSC and MSC behavior within the bone marrow niche to elucidate how a dysregulated bone marrow niche contributes to MPNs’ disease progression. Our findings highlight the significant impact of ECM components, cell-cell and cell-matrix interactions, and mechanosensitive structures on HSC behavior. We also explored the primary cilium’s role as a chemo- and mechanosensory organelle in the bone marrow niche, particularly its involvement in critical signaling pathways such as Hh and Wnt, which are essential for regulating hematopoiesis. Although further studies are needed to specifically investigate molecular mechanisms through which PC and other mechanosensors are involved in HSC homeostasis regulation, may alter HSC cycling, self-renewal, homing, and lineage potential, we offer a tempting speculation on the potentially leading role of primary cilium expressed on MSCs in bone marrow disarranged mechanobiology following a severe impairment during MPN. In that regard, we underscore the disruptions in bone marrow architecture and cellular composition associated with MPNs which expose HSCs to a distinct mechanical environment with potential implications for hematologic dysfunction and disease progression.

A better understanding of the wide and heterogeneous array of mechanobiological signals that govern bone marrow homeostasis and pathophysiology will indeed grant a significant step forward in hematology translational research.

Furthermore, future investigations into primary cilium formation and features in BM-MSC subpopulations, such as Lepr^+^ MSCs and associated signaling pathways (e.g., Hh), could enhance our comprehension of fibrosis in MPNs. These studies will also be key to identifying potential therapeutic targets to improve patient outcomes. Further evaluations in HSC cultures will help to elucidate the correlation between MSCs and HSCs in disease progression. Moreover, recent discoveries linking anticancer drug resistance with primary ciliary dynamics highlight the primary cilium as a critical and innovative target for overcoming drug resistance.

## Figures and Tables

**Figure 1 ijms-25-08860-f001:**
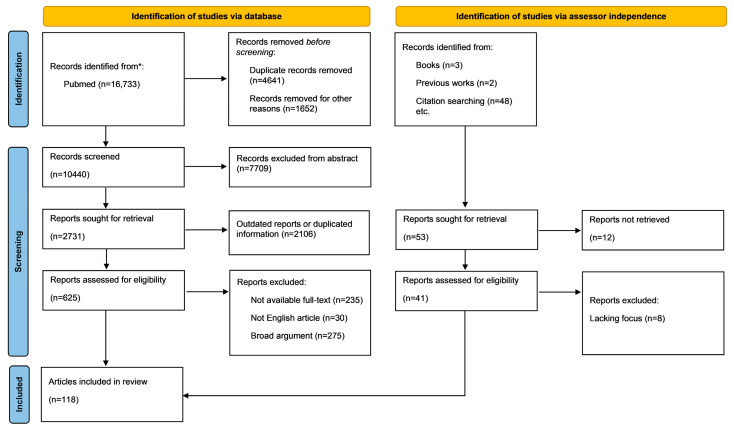
PRISMA 2020 flow diagram for new systematic reviews, which included searches of databases, registers, and other sources * [17]. This work is licensed under CC BY 4.0. To view a copy of this license, visit https://creativecommons.org/licensed/by/4.0/ (accessed on 15 June 2024).

**Figure 2 ijms-25-08860-f002:**
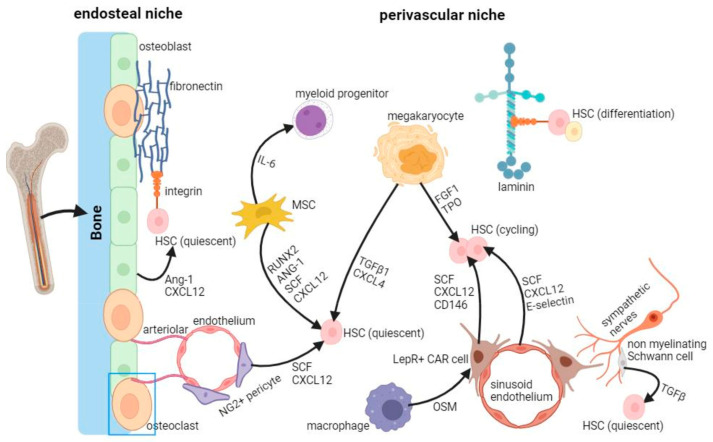
Niches in the bone marrow. Adult hematopoietic stem cells (HSCs) are organized within the bone marrow microenvironment, in which two main different cellular niches are recognized: the endosteal and the perivascular niche. As depicted on the left of this Figure, the endosteal niche plays a key role in maintaining HSCs quiescence, while the perivascular niche (on the right), activates the cell cycle and promotes proliferation. Both biochemical and biomechanical factors contribute together to shape the environment. LepR, leptin recep-tor-expressing; CAR cell, cxcl12- abundant reticular cell; MSC, mesenchymal stromal cell; SCF, stem cell factor; FGF1, fibroblast growth factor 1; TGFβ, transforming growth factor β; NG2, neuron-glial 2; OSM, oncostatin M (created by Biorender.com).

**Figure 3 ijms-25-08860-f003:**
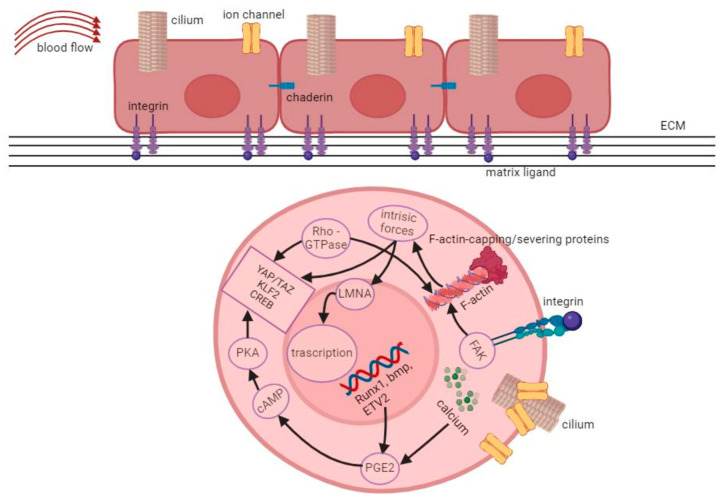
Biomechanical signals and molecular pathways in hematopoietic stem cells (HSCs). Simplified representation of how external mechanical stimuli interact with mechanosensors like ion channels, adhesion receptor-ligand bonds, cytoskeletons, and primary cilia. Mechanosensing does impact several translational events (summarized in this Figure), although all of the molecular pathways activated are still not known. The role of the junctional interfaces is important to transmit internal forces, triggering cellular mechanical responses. See the text for a more detailed explanation (created using Biorender.com).

**Figure 4 ijms-25-08860-f004:**
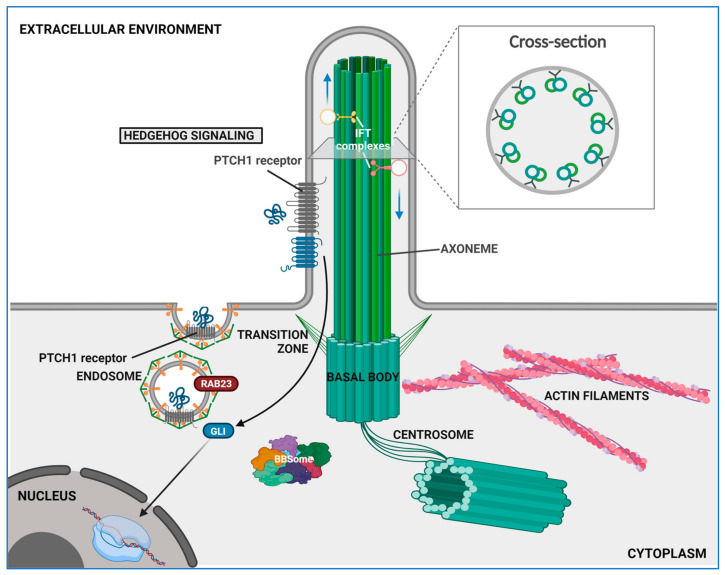
Primary cilium structure. Schematic representation of the primary cilium extending from the apical surface of cells, featuring a central core made of nine microtubule doublets (axoneme) that originate from the basal body, a modified centrosomal mother centriole. The transition zone (TZ) includes Y-shaped links that connect the axonemal microtubules to the ciliary membrane, helping to compartmentalize the organelle. Proteins and other cargos are moved anterogradely from the basal body to the tip of the axoneme by the intraflagellar transport (IFT)-B complex and kinesin motor protein, while the IFT-A complex and dynein motor protein handle retrograde transport from the tip to the basal body. The basal body is associated with the BBSome, a seven-protein complex crucial for ciliogenesis and ciliary trafficking. The ciliary membrane is enriched with specialized lipids, proteins, and receptors such as the Patched 1 (PTCH1) receptor and the Smoothened (SMO) co-receptor, which interact with various Hedgehog (HH) ligands to regulate signaling pathways (e.g., Hedgehog signaling). Refer to the text for more details. Modified using BioRender from Tiberio et al. [49].

**Table 1 ijms-25-08860-t001:** Criteria for MPN classification.

Polycythemia Vera	Essential Thrombocythemia	Primary Myelofibrosis
**Major criteria**		
- HGB > 16.5 g/dL (men), >16 g/dL (women). - HCT > 49% (man), >48% (women). - Increased erythrocyte mass (>25% more than the reference value). - BM biopsy showing hypercellularity with erythroid, granulocytic, and megakaryocytic proliferation and with pleomorphic mature megakaryocytes. - Presence of *JAK2*V617F or exon 12 mutation.	- Platelet ≥ 450 × 10^9^/L. - BM biopsy showing proliferation mainly of the megakaryocytic lineage with increased numbers and sizes, mature megakaryocytes with hyper-lobulated nuclei. No significant increase or left shift in granulopoiesis or erythropoiesis and very rarely minor increase in reticulin fibers. - Absence of diagnostic criteria for other myeloid neoplasms or myelodysplastic syndromes. - Presence of *JAK2*, *CALR*, or *MPL* mutation.	- Megakaryocyte proliferation with atypia and reticulin and/or collagen fibrosis. In the absence of reticulin fibrosis, the presence of granulocyte proliferation and a reduction in the erythrocyte lineage can lead to a prefibrotic phase. - Absence of diagnostic criteria for other neoplasms of the myeloid lineage. - Presence of *JAK2*V617F mutation or another clonal marker, and if absent, exclusion of reactive fibrosis.
**Minor criteria**		
- Serum EPO levels below the reference value.	- Presence of clonal markers and no evidence of reactive thrombocytosis.	- LDH increased. - Anemia. - Palpable splenomegaly. - Leukocytosis ≥ 11 × 10^9^/L. - Leukoerythroblastosis.
**Criteria for diagnosis**		
For the diagnosis of PV, three major criteria are required or, alternatively, two major criteria and the minor criterion.	For the diagnosis of ET, all four major criteria are required or the first three major criteria and the minor criterion.	For the diagnosis of PMF, patients must present all three major criteria, and at least one minor criterion.

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
