# Peer review of "Mechanobiology and Primary Cilium in the Pathophysiology of Bone Marrow Myeloproliferative Diseases"

_ijms, 2024, doi:10.3390/ijms25168860_

Round 1

Reviewer 1 Report

Comments and Suggestions for Authors

In this manuscript, Tiberio et al. summarized bone marrow niche and HSC biology, and how mechanical signals and primary cilium could regulate HSC homeostasis and contribute to MPNs. The review is well-organized with good clarity. I have a few comments listed below regarding to the revision process.

1. The title emphasizes pathophysiology of MPNs which was not the main focus of the whole paper. “Mechanical cues” were not found as repeating key words. I suggest to edit the title/body so that they are consistent.

2. Figure 1 contains a lot of information. But if the message is important, why not elaborate bone marrow niche in the text rather than only in figure legend? Same with figure 2.

3. The conclusion is too brief and can be stronger. Maintain the focus established in the introduction; summarize major research contributions to the field with clear point of view; point out major flaws/ gaps in research; highlight potential future studies, etc.

Author Response

COMMENT 1. The title emphasizes pathophysiology of MPNs which was not the main focus of the whole paper. “Mechanical cues” were not found as repeating key words. I suggest to edit the title/body so that they are consistent.

AUTHORS' REPLY: We thank Reviewer#1 for the comment. We have edited the title of the manuscript using “mechanobiology” instead of “mechanical cues” (line 2).

COMMENT 2: Figure 1 contains a lot of information. But if the message is important, why not elaborate bone marrow niche in the text rather than only in figure legend? Same with figure 2.

AUTHORS' REPLY: We thank the Reviewer for this comment and for giving us the possibility to better describe bone marrow niche in the text. We have changed the text (lines 157-163 and lines 217-220) accordingly and simplified the legends (see the revised Figure 1 and 2 legends).

COMMENT 3: The conclusion is too brief and can be stronger. Maintain the focus established in the introduction; summarize major research contributions to the field with clear point of view; point out major flaws/ gaps in research; highlight potential future studies, etc.

AUTHORS' REPLY: We thank the reviewer#1 for the constructive feedback. We have revised the conclusion to incorporate the suggested points (see text – 6. Conclusion- for details; lines 1022-1050).

Reviewer 2 Report

Comments and Suggestions for Authors

In the present narrative review, the authors examine the mechanobiology within the bone marrow niche, emphasizing the role of mechanical cues and the primary cilium in the pathophysiology of MPNs, the influence of extracellular matrix components, cell-cell and cell-matrix interactions, and mechanosensitive structures on hematopoietic stem cell (HSC) behavior and disease progression, as well as the potential implications of the primary cilium as a chemo- and mechanosensory organelle in bone marrow cells.

The manuscript is overall well-written and scientifically sounds and warrants publication following some minor revisions.

Comments:

- please revise the diagnostic criteria for MPNs to match the most recent ICC/WHO guidelines, including the criteria for secondary MF

- does oxidative stress also play a role in the interaction between the bone marrow niche and the cillium?

- please also discuss the relevance of extracellular vesicles, microparticles, and the potential role of liquid biopsy in MPNs

- I think you should add a methodology section explaining how selection of the included studies was made and which databases were consulted

- references are not formatted in the MDPI style (American Chemical society) 

Author Response

COMMENT 1: Please revise the diagnostic criteria for MPNs to match the most recent ICC/WHO guidelines, including the criteria for secondary MF

AUTHORS' REPLY: We thank the Reviewer, we have replaced Table 1 with new tables according to ICC/WHO guidelines (see lines 51-54), as supplementary material (S1-S6).

COMMENT 2: Does oxidative stress also play a role in the interaction between the bone marrow niche and the cilium? 

AUTHORS' REPLY: We thank reviewer#2 for the suggestion. The manuscript has been revised and improved considering this aspect (see text for details - 4.3 The functional significance of the primary cilium in the bone marrow niche; lines 917-957).

COMMENT 3: please also discuss the relevance of extracellular vesicles, microparticles, and the potential role of liquid biopsy in MPNs

AUTHORS' REPLY: Thank you for your comment. However, the relevance of extracellular vesicles, microparticles, and the potential role of liquid biopsy in MPNs falls outside the primary focus of our manuscript. Our study specifically addresses the mechanobiology within the bone marrow niche, emphasizing the role of mechanical cues and the primary cilium in the pathophysiology of MPNs, and integrating these additional topics would diverge from the core objectives and scope of our research, since this is not directly related to the specific aims and findings presented in our manuscript. We appreciate your understanding and the opportunity to clarify this aspect of our work.

COMMENT 4: I think you should add a methodology section explaining how selection of the included studies was made, and which databases were consulted

AUTHORS' REPLY: We thank reviewer#2 for the suggestion. We added a methodology section on this aspect (see text for details; lines 131-147 and Table 2)

COMMENT 5: references are not formatted in the MDPI style (American Chemical society)

AUTHORS' REPLY: The reference list has been updated and provided in the appropriate format.

Reviewer 3 Report

Comments and Suggestions for Authors

The authors have provided a valuable resource in this comprehensive review on the subject of how mechanical forces affect hematopoiesis, with a special focus on the primary cilium. An enormous amount of information is gathered succinctly, with an appropriate number of references, and the Figures are helpful. Only one error stands out: the mention of "triplets" on line 289, whereas "doublets" is presumably intended. Otherwise, I have only optional suggestions for improvement:

1) The connection to MPD is the weakest part of the review. That is apparent from the brevity of section 4 alone, which implies that not much is really known. It is readily plausible that the mechanobiology of hematopoiesis is disrupted in MPDs, especially with myelofibrosis, but the review is not clear as to whether that is cause or effect of the MPD. Is there any better evidence on this point, such as PMID: 20305640, in which a genetic defect in MSCs led to myelodysplasia and leukemia. In other words, is there an example in which disruption of the primary cilium and/or mechanotransduction of HSCs led to MPD-like disorders?

2) In keeping with restraints on length, the review is long on conclusions and short on the studies and methods on which they are based. However, it would add interest and comprehensibility to provide some information about how the primary cilium and mechanotransduction have been studied. E.g., how is fluid flow in the bone marrow measured? What are new techniques that the authors expect to be insight-generating?

3) Given the authors' expertise in this field, the reader would benefit from knowing their thoughts, even if purely speculative, on how understanding primary cilium and mechanotransduction can be translated into therapy.

Author Response

COMMENT 1: The authors have provided a valuable resource in this comprehensive review on the subject of how mechanical forces affect hematopoiesis, with a special focus on the primary cilium. An enormous amount of information is gathered succinctly, with an appropriate number of references, and the Figures are helpful. Only one error stands out: the mention of "triplets" on line 289, whereas "doublets" is presumably intended. Otherwise, I have only optional suggestions for improvement:

AUTHORS' REPLY: We thank reviewer#3 for appreciating the manuscript and for pointing out the error in our manuscript. We have corrected the term "triplets" to "doublets" on line 391 as suggested.

COMMENT 2: The connection to MPD is the weakest part of the review. That is apparent from the brevity of section 4 alone, which implies that not much is really known. It is readily plausible that the mechanobiology of hematopoiesis is disrupted in MPDs, especially with myelofibrosis, but the review is not clear as to whether that is cause or effect of the MPD. Is there any better evidence on this point, such as PMID: 20305640, in which a genetic defect in MSCs led to myelodysplasia and leukemia. In other words, is there an example in which disruption of the primary cilium and/or mechanotransduction of HSCs led to MPD-like disorders?

AUTHORS' REPLY: We thank the reviewer for the comment. Clearly, not much is known about the link between disruption of mechanotransduction and Myeloproliferative disorsorders. In particular, we think that Myeloproliferative neoplasms can certainly have an impact on mechanotransduction and signaling in the bone marrow microenvironment.

On the other hand there is only manuscript, to our knowledge, in which it is mentioned a gene involved in ciliogenesis who has also been shown to be partner of an oncogenic fusion protein (https://pubmed.ncbi.nlm.nih.gov/23554904/) FOP (FGFR1 Oncogene Partner) is a centriolar satellite cargo protein involved in ciliogenesis  and FOP-FGFR1, fusion causes myeloproliferative neoplasm. However, the functional significance of centrosome-kinase fusions in myeloproliferative neoplasms is not well understood, and it has been hypothesized that aberrant kinase localization is a factor in the disease phenotype.

We have changed the text to include this information (lines 987-993) and revised the text throughout the paragraph.

COMMENT 3: In keeping with restraints on length, the review is long on conclusions and short on the studies and methods on which they are based. However, it would add interest and comprehensibility to provide some information about how the primary cilium and mechanotransduction have been studied. E.g., how is fluid flow in the bone arrow measured? What are new techniques that the authors expect to be insight-generating?

AUTHORS' REPLY: We thank reviewer#2 for the suggestion. We improved the section - 4.1 Structural and functional overview- reporting recent data collected from literature studies (see text for details; lines 427-442).

COMMENT 4: Given the authors' expertise in this field, the reader would benefit from knowing their thoughts, even if purely speculative, on how understanding primary cilium and mechanotransduction can be translated into therapy.

AUTHORS' REPLY: thank the reviewer for the constructive comment. We detailed this aspect in section - 6. Conclusion (lines 1043-1050).